# Three Historic Tide Gauge Records from Svalbard

Philip L. Woodworth [1], Thorkild Aarup [2]

[1]National Oceanography Centre, Joseph Proudman Building, 6 Brownlow Street, Liverpool, L3 5DA, United Kingdom (retired)
[2]Intergovernmental Oceanographic Commission, UNESCO, 7 Place de Fontenoy, 75352 Paris, Cedex 07 SP, France (retired)

*Correspondence to*: Philip L. Woodworth (plw@noc.ac.uk)

**Abstract.** Three historic tide gauge records from the Arctic archipelago of Svalbard have been recently converted from tabulations more than one century old into computer files. The records are found to be good quality and capable of being used in modern tidal analysis. The analyses confirm the findings on tidal constants by previous researchers and demonstrate how little non-tidal variability in sea level there was at these times. One of the tide gauges used was a crude contraption of a design not used before or since. Nevertheless, it appears to have worked well and so deserves to be better known.

## 1 Introduction

This report discusses three tide gauge records from Svalbard that were obtained during the end of the 19th century (Table 1). The novelty of the records stems from their historical importance, for example within studies of whether local tides have changed during the past century; the geographical locations of the gauges, records from the high Arctic being very sparse; and, for one of the gauges, its unusual construction which (to our knowledge) has never been adopted before or since. The records are in the form of half-hourly or hourly heights of sea level. They have been subjected to modern tidal analysis with the main tidal constants so obtained for two of them compared to those existing in tidal databanks. The constants have also been compared to those of a state-of-the-art numerical tidal model.

The following report describes the sources of the records, the tide gauge equipment used in each case, and results of the tidal analyses.

Table 1. Locations of tide gauges with records analysed in this report.

| Station | Longitude (°E) | Latitude (°N) | Measurement Span | Sampling | Notes |
|---|---|---|---|---|---|
|  |  |  |  |  |  |
| Bay of Treurenberg/Sorgfjord | 16.840 | 79.919 | 28 March – 12 July 1900 | Hourly | 1 |
| Port Virgo/Danskegat | 10.870 | 79.723 | 8 June – 18 July 1897 | Hourly | 2 |
| Mossel Bay/Mosselbukta | 16.067 | 79.883 | 20 October – 26 November 1872<br>18 February – 24 April 1873 | Half-hourly<br>Hourly | 3 |

Notes:

1. Coordinates shown are from Google Earth. CG05 gives the longitude as 16.858 °E. The ATT (station 891) shows the

coordinates as 16° 50' E, 79° 55' N (16.833 °E, 79.917 °N).

2. Coordinates shown are from CG05 as supplied by Nils Strindberg (see the text). The ATT (station 895) shows the coordinates as 10° 52' E, 79° 43' N (10.867 °E, 79.717 °N).

3. Coordinates shown are as reported by Wijkander (1889). The ATT (station 892) shows the same coordinates 16° 04' E, 79° 53' N.

## 2 The Three Tide Gauge Records

### 2.1 Data from 1900 from Sorgfjord

The first record discussed was acquired during the Swedish-Russian Arc-of-Meridian expedition to the Arctic archipelago of Svalbard (Fig. 1a). During 1899-1902 an extensive triangulation campaign was undertaken there in order to measure the length of a meridian arc (Conway, 1903; Norwegian Polar Institute, 2025a). Such exercises have been undertaken in other parts of the world since ancient times, with the object of determining the radius and shape of the Earth and, specifically in this high-latitude case, the extent to which the spherical Earth is flattened at the poles (Wikipedia, 2025a). The expedition was said to have been one of largest projects of its kind in history (Shabalina and Kazakova, 2022; Kyzyurova, 2023). The Russians took responsibility for the southern measurements on Spitsbergen, the largest of the Svalbard islands, while the Swedes performed measurements in northern Spitsbergen, extending as far as Little Table Island off the north coast of Nordaustlandet, the second-largest island in the archipelago. The Swedish measurements were made from a winter base on the eastern side of Sorgfjord (Bay of Treurenberg) on the north coast of Spitsbergen (Fig. 1b). The location of the base is called Crozier Point, named after Francis Crozier, a lieutenant during William Parry's Arctic expedition in 1827, who died in 1848 while second in command of the Franklin expedition.

The campaign made use of a chain of triangulation points on mountain tops across Svalbard, forming 22 triangles and spanning more than 4 degrees of latitude. A baseline was established at the Swedish station using the so-called Jäderin apparatus, consisting of approximately 25 m of Guillaume metal (steel-nickel alloy) wire stretched at fixed tension over a series of tripods.

Edvard Jäderin is a name which crops up frequently in this story. He was a Swedish geodesist who led the Swedish preparatory expedition to Svalbard in 1898 and was leader of the Arc-of-Meridian wintering party in Sorgfjord in 1899-1900 (Wikipedia, 2025b). He seems to have constructed much of the equipment used by the Swedes.

As in any triangulation campaign, the height and horizontal coordinates of a point in the network are calculated in an iterative fashion by considering a triangle defined by the known length of a section of ground near to the point, and two angles measured from the ends of the section to that point. The lengths of the other two sides of the triangle then become known and the exercise moves on to the next point and triangle. Consequently, if the height above mean sea level (MSL) of the baseline is known, then so are the heights of each point in the network. Analysis of data from a tide gauge at the Swedish station provided that essential MSL information, with the height of the baseline determined by knowing the MSL, and also the height-difference between gauge and baseline obtained using conventional levelling. That levelling was conducted by Jäderin.

The tide gauge record obtained consists of hourly values of sea level spanning more than 3 months (28 March – 12 July 1900) with only occasional interruptions of some hours when the missing readings were inferred in the subsequent analysis by

interpolation. These values are tabulated in Appendix 1 of the report of Carlheim-Gyllensköld (1905, hereafter CG05). Carlheim-Gyllensköld seems to have been another interesting character associated with the expedition with special interests in the Earth's magnetic field; a biography of him can be found in Ljungdahl (1935).

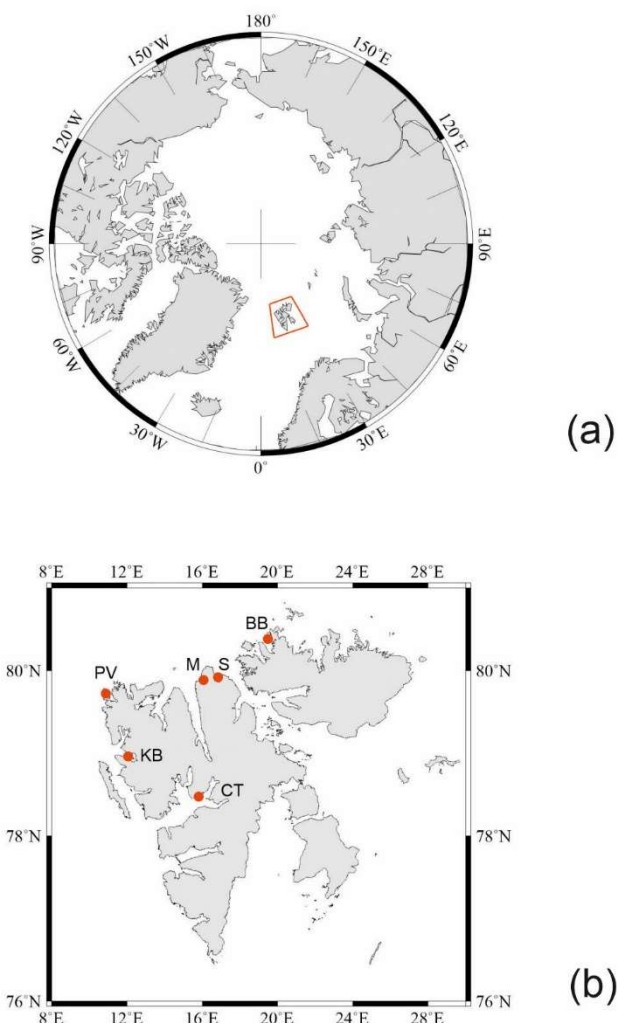

**Figure 1: (a) Map of the Arctic Ocean showing the location of the Svalbard archipelago (red box), (b) Map of Svalbard showing red dots for locations mentioned in the text: Sorgfjord (S), Mossel Bay (M), Port Virgo (PV), Brandy Bay (BB), Cap Thordsen (CT) and Kings Bay (KB).**

The tide gauge was installed on 26 March 1900, the first couple of days of data being of doubtful quality. A schematic of the unusual equipment is given in CG05, reproduced here as Fig. 2, while two grainy photographs from CG05 are shown in Fig. 3(a,b).

From right to left, the end of a strong wire 'aixd' (approximate diameter 2.25 mm) was anchored at the foot of a post 'd' buried
in the ground. The wire then crossed through a hasp 'x' fixed to a large post 'c', the foot of which was weighted down by stones. The hasp will have held the wire strongly in place. The wire then passed at 'i' above the tide gauge recorder 'm' at position 'b', and extended to a pulley 'y' supported on a solidly constructed tripod 'a', frozen into the ice in the bay. On the end of the wire below the pulley was a large block of stone weighing 75-100 kg. The tripod 'a' was located 80-85 m from the shore so that the vertical movement of the ice was sheltered from the influence of the broken ice near the land. Assumptions
behind the arrangement will have included that the frozen ice did not allow the tripod to drift, or for the tripod to sink into the ice, or for the wire to be affected by wind or thermal forces.

The idea was that, because the ratio of distances xi/xy was 1/17.22, then the vertical motion of the ice due to the tide under the tripod will have resulted in corresponding motion at 'i' scaled by 17.22. A thin 'red copper' wire attached to the main wire at
'i' entered vertically into the tide gauge apparatus 'm' and was attached to a slide with a crayon which made a trace on a clock-driven chart recorder; CG05 shows an example of one such chart. The chart recorder could work for 8 days, charts being replaced every Monday at 10 am. Each chart measured 410 by 170 mm and the cylinder was said to turn at a rate of 2.3 mm per hour (which sounds rather slowly in retrospect). The time used for the observations was local mean time during the astronomical (not civil) day.
In order to provide datum control to the tide gauge measurements, and thereby control the zero of the traces on the charts, a levelling sight was installed next to the tripod on the ice from which measurements were made at regular intervals using a levelling telescope. CG05 contains more details of this procedure. A method was also devised to relate the tide gauge readings to benchmarks on land by making 'dipping' measurements through a hole in the ice near to the tripod 'a' and using conventional
levelling to relate the dipping rod to a mark on the block on which the recorder 'm' was installed. These measurements were made around high and low tides when the water level would have been changing only gradually and the results of each reading were noted on the paper chart.

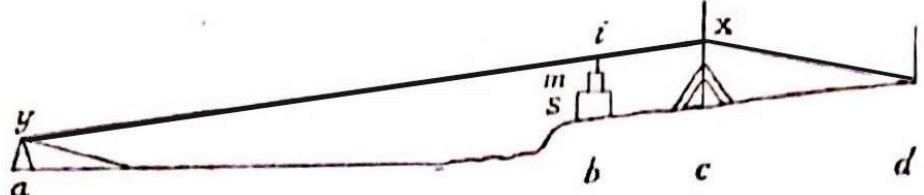

**Figure 2: Schematic from CG05 of the tide gauge used at Sorgfjord in 1900. A wire 'aixd' passes over a tripod at point 'a' on floating ice approximately 80m from the shore, and points 'b', 'c' and 'd' on land. A thinner wire extends vertically from 'i' on the main wire through a container and is attached to a slide with a crayon that writes on a chart recorder 'm'. As a result, variations in the level of the ice at 'a' are reproduced on the chart recorder 'm' scaled by 17.22 to 1. (Source: Carlheim-Gyllensköld, 1905).**


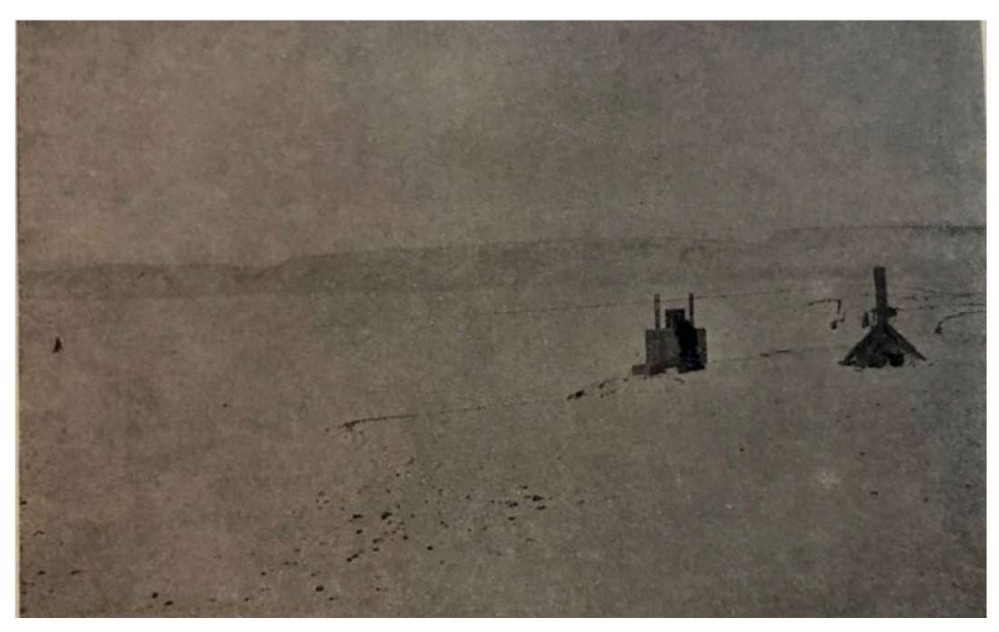

(a)

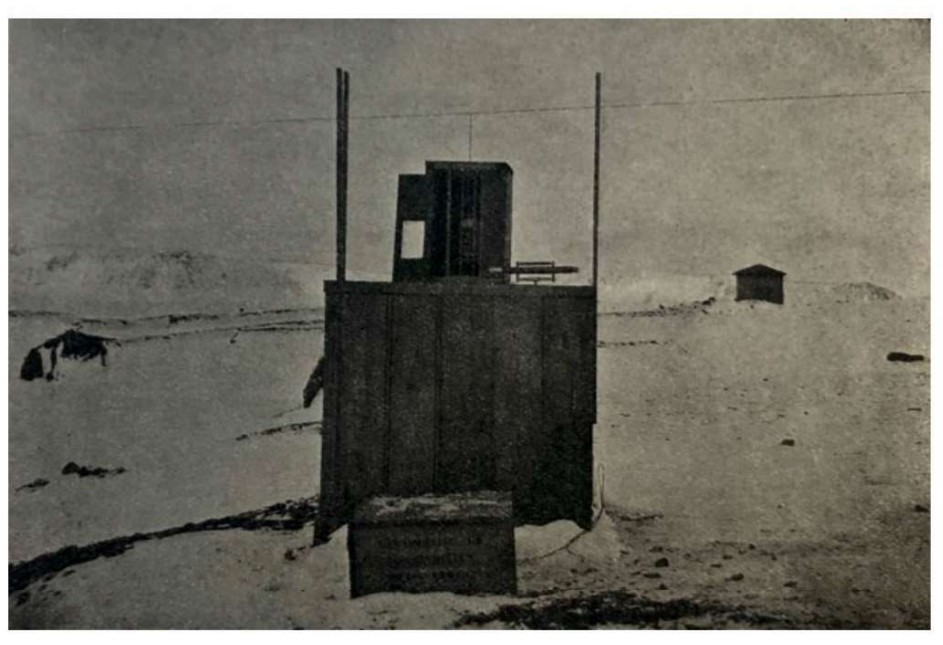

(b)

**Figure 3: (a) Photograph from CG05 of the tide gauge arrangement shown schematically in Figure 2, showing the post 'c' on the right, tide recorder arrangement 'b' in the centre and the tripod on the floating ice 'a' on the left. The wire passing over 'b' and 'c' and extending out to 'a' can just be seen. (b) A close view of the tide recorder at 'b', the wire passing overhead can just be seen, as can a thinner vertical wire extending down from the main wire into the chart recorder apparatus. (Source: Carlheim-Gyllensköld, 1905).**

The resulting measurements made by Jäderin spanned 105 consecutive days. Some interruptions of different lengths occurred during 2 days in April, 8 in May, 5 in June and 2 in July. In these cases, estimated curves were drawn on the charts with the numbers so interpolated printed in italics in Appendix 1 of CG05. Carlheim-Gyllensköld emphasised that the apparatus was delicate and depended on a number of conditions being met. One was that it was necessary for the tension of the wire to be constant, with no other forces acting on it (such as winds) other than the weights at the tripod end. In addition, it was essential for the crayon to move without resistance and for the clock to be well maintained. Otherwise, various errors could affect the data, and CG05 discusses in detail what the consequences of them might be.

Nevertheless, in spite of the somewhat crude equipment, Jäderin is to be congratulated for constructing at short notice something which worked for an extended period in difficult conditions. CG05 says that it had been intended for the expedition to use a modern gauge made by Petrelius but that the equipment had not been made available by the time the expedition departed for Svalbard.

As an aside, this is a reference to the Finnish geodesist Alfred Gustav Petrelius (Wikipedia, 2025c), who patented a type of tide gauge that had been demonstrated to work in winter conditions at several locations in Finland and was not expensive; the Swedish expedition must have tried to acquire one. This was another unusual tide gauge, being essentially a conventional float gauge but with a double U-bend, part-filled with mercury and part with oil to avoid the mercury mixing with dirty sea water. A patent for the gauge may be found at Petrelius (1900).

Table 2 shows the harmonic constants for the main components of the tide as computed by Carlheim-Gyllensköld. These comprise the amplitudes and phase lags of the two main semidiurnal (M2 and S2) and diurnal (K1 and O1) constituents. CG05 includes several pages explaining how the tide can be parameterised as a set of waves and introduces Carlheim-Gyllensköld's own computational method of tidal harmonic analysis. Modestly, he admits that his own method is imperfect and that his values may be modified following new calculations. In addition, he states correctly that his values may be affected by local meteorological influences and technical considerations, which is the case for any computational method. Pugh and Woodworth (2014) may be consulted for how such tidal parameters are determined nowadays from analysis of tide gauge records.

In fact, the constants computed by Carlheim-Gyllensköld appear consistent with later analyses of the data (except perhaps for the phase lag of the small O1 constituent), such as those shown for Sorgfjord as station 891 in the Admiralty Tide Tables

(ATT, 2009), and included in the present Table 2. The records of the US National Oceanic and Atmospheric Administration (NOAA) show that these ATT values had been documented by them in 1937 or earlier, and it had been assumed that they were the result of a reanalysis of the data in the mid-1930s. However, we now believe that they were computed far earlier, by the US Coast and Geodetic Survey (C&GS) for the publication of Harris (1911) on Arctic Tides (see page 42, station 13) and have remained the 'official' ATT values to the present day.

In fact, Harris (1911) also included the CG05 values on his page 42 as station 24 but failed to realise that the CG05 phase lags were Greenwich ones whereas the C&GS ones were local phase lags (usually denoted by the Greek letter kappa in tidal literature), hence his comment that the CG05 values were 'evidently erroneous'. We believe this is not the case, as can be seen in the present Table 2 when both sets are expressed as Greenwich phase lags.

Table 2: Amplitudes (cm) and Greenwich phase lags (deg) resulting from tidal analysis of the hourly heights at Sorgfjord, Spitsbergen in 1900, as reported in CG05, as given in the ATT, and as computed in the present analysis. The corresponding parameters from the FES2022b model are also shown.

| Constituent | CG05 | ATT station 891 (Sorgfjord) (*) | Present Analysis | FES2022b |
|---|---|---|---|---|
| M2 Amplitude | 28.0 | 28 | 27.8 | 29.04 |
| Phase lag | 64.6 | 65 | 65.0 | 58.07 |
| S2 Amplitude | 10.1 | 11 | 10.2 | 10.29 |
| Phase lag | 116.9 | 116 | 116.5 | 107.92 |
| K1 Amplitude | 8.3 | 7 | 7.4 | 6.98 |
| Phase lag | 254.7 | 253 | 254.5 | 257.48 |
| O1 Amplitude | 2.6 | 2 | 2.1 | 1.68 |
| Phase lag | 137.5 | 53 | 54.0 | 62.70 |

* The ATT lists phase lags in time zone -0100 which have been converted here to GMT.

Our own calculations are also shown in Table 2. These were obtained using the tidal software of Bell et al. (1998), in which fits to the data are made using 27 independent and 8 dependent harmonic constants, as is appropriate for short records. Our findings agree closely with those of both CG05 and C&GS/ATT.

Because of the problems reported with the gauge at the end of March, our analysis determines the astronomical tide using data from the start of April onwards. The residuals of the analysis are shown in Fig. 4. They show little evidence for any significant non-tidal sea level variability, such as occurs during storm surges, these of course being spring-to-summer months. The only problems seem to have been at the end of March, as mentioned, and on 29 May and 17-18 June when the residuals are tidal in character. This suggests an error in either chart recording or digitisation. However, these two short periods will have no effect on our overall determination of harmonic tidal constants.

In addition to the ice-surface gauge installed by Jäderin, measurements had been made at Sorgfjord with what seem to have been float gauges during the short preceding autumn of 1899. (For a description of a stilling-well float gauge see Pugh and Woodworth, 2014). One of these devices was placed in a small recess in the cliff to the south of the magnetic observatory. A plank fixed in a horizontal position served to support another plank fixed vertically with its end on the sea bed. A wooden float

pierced with a large hole enveloped the vertical plank and could glide without friction along it, thereby indicating the water level. The vertical plank was graduated and the float was equipped with an index with which one could measure its position on the scale. This wooden float was gradually saturated with sea water and its results were not very satisfactory. A second stilling-well type of device, said to be constructed by Westman, was established in better conditions. Inside the well was a float consisting of a tin can (as used for preserved food and resoldered) extending upwards into a thin stem. On the upper extremity of the well was a tin plate with a circular hole 1 cm in diameter that let the stem pass through without friction. The stem was graduated in centimetres and readings were made at the upper surface of the tin plate; fractions of cm were obtained by estimation. During the night of 13/14 September the new gauge was destroyed by swell. On 16 September 1899, the zero of the second gauge was determined with respect to benchmarks established near to the observatory. (Harmonic constants derived from this very small set of data are included in Harris (1911), page 42 station 23).

Incidentally, the Westman mentioned must have been Jonas Westman, a Swedish meteorologist and teacher and a member of the Arc-of-Meridian Expedition during which he wintered in Sorgfjord in 1899-1900. A mountain in Svalbard is named after him (Chernouss and Sandahl, 2008; Norwegian Polar Institute, 2025b; Wikipedia, 2025d).

CG05 contains much more information of how both temporary gauges operated over a short period, with a week of overlapping operation in August, with which the height of the benchmarks on land could have been roughly related to MSL prior to the Arc-of-Meridian expedition. The fine details need not concern us here except to note that the people involved clearly knew what they were doing and made useful measurements with the most basic equipment in what must have been difficult conditions.

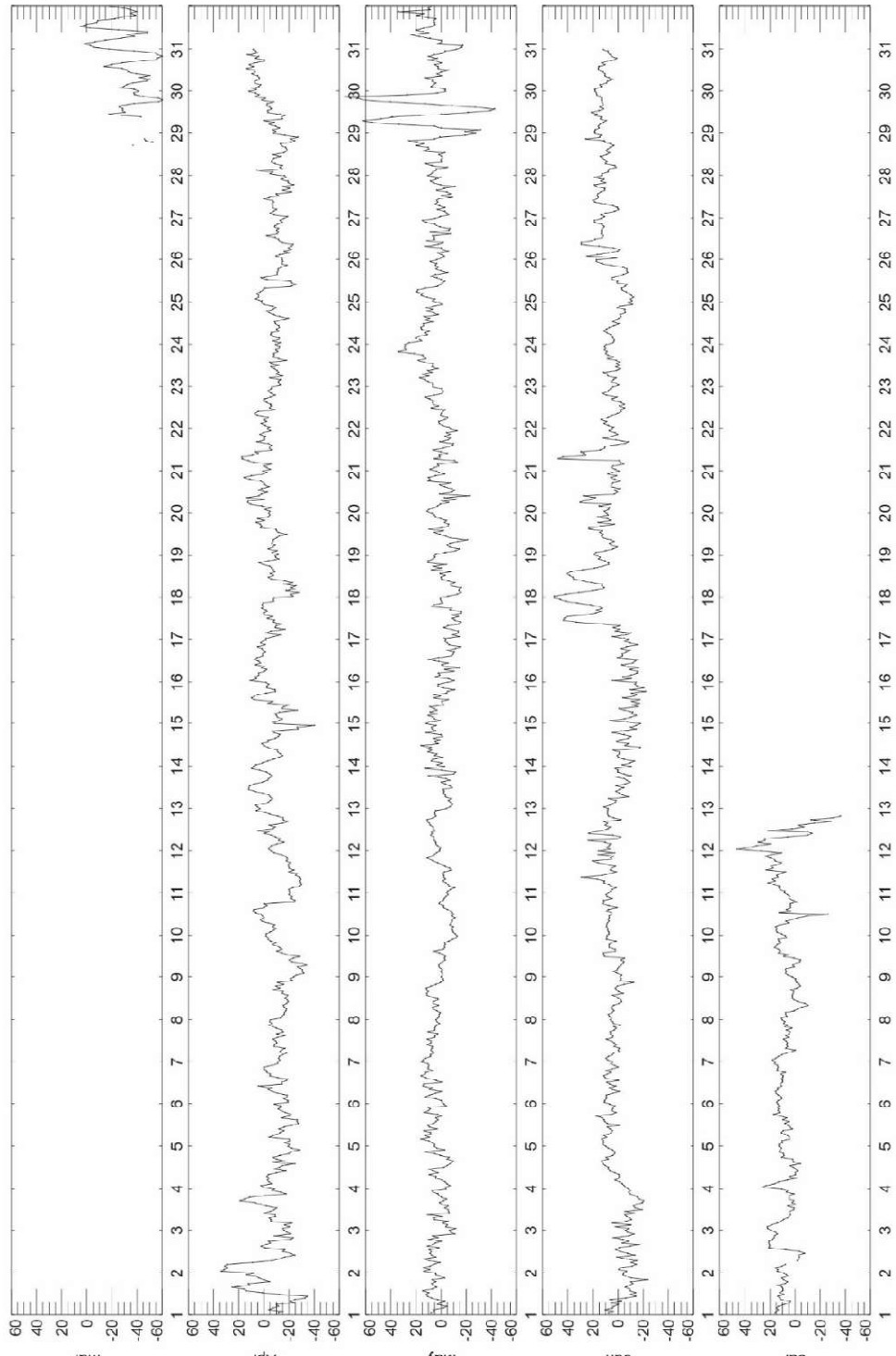

**Figure 4: Residuals of the tidal analysis for the data from Sorgfjord in 1900. Units are cm.**

**2.2 Data from 1897 from Port Virgo**

The second record discussed in CG05 is of shorter duration, commencing at 2300 hours on 8 June until 0800 hours on 18 July 1897, at Port Virgo, Île des Danois (Danskegat) on the northwest coast of Spitsbergen (Fig. 1b). The data take the form of hourly values of sea level. These measurements were made during the second expedition to Svalbard of Salomon August Andrée, in which he made an unsuccessful attempt to reach the North Pole by balloon (Wilkinson, 2012). The ship belonging to the expedition was the gunboat Svensksund. Measurements were made using a simple tide board (tide pole) constructed on the shore of the Île des Danois by a number of the ship's officers and sailors who shared the responsibility. A book containing tabulations of the observations was given to Carlheim-Gyllensköld by one of the sailors called Dahlgren who later took part in the Arc-of-Meridian mission of reconnaissance to Svalbard in 1898.

It is not clear why these measurements are included in CG05, which after all was a report on the Arc-of-Meridian expedition itself. One assumes that Carlheim-Gyllensköld considered them to provide interesting background information. Anyway, they are of interest to us today from a tidal perspective.

Several interruptions occurred in the record and in these cases a graphical method was made to complete the gaps; the interpolated values are printed in italics in the table on page 56 of CG05. The table says that the time employed was local mean time, whereas the text says it was either the local apparent time (temps vrai) as reported by a person called Carlson, or mean time at Christiania (Oslo) suggested by someone else. The text then says that the equation of time (the difference between apparent and mean time) will be of no importance.

Carlheim-Gyllensköld performed his own tidal analysis of these data and concluded rightly that a longer period of data is needed for more definitive results. However, his findings shown in Table 3 appear to be reliable as shown below.

For our own analysis of the data, a first problem that arose was the confusion over what sort of time was used for the observations. This is potentially important, as is knowledge of the longitude at which the measurements were made, so that one can accurately compute harmonic constants with Greenwich phase lags. This involves adjusting the recorded times to Greenwich Mean Time (GMT). However, in fact, these different timing and longitude uncertainties have been found to have little importance for present purposes.

As for timing uncertainty, mean and apparent times are identical to within a few minutes during June and July, giving an uncertainty of about 0.05 hour, which corresponds to 0.75 and 1.5 deg uncertainty in phase lags for diurnal and semidiurnal

tides respectively. As for longitude uncertainty, the text of CG05 says that the measurements were made at 10° 52' 12'' E, 79° 43' 24'' N, as reported by Strindberg (1897). (Nils Strindberg was a photographer who died during the Andrée expedition, Wikipedia 2025e.) This is very close to the coordinates for the base on the Île des Danois in Google Earth (GE), so we believe them, and we have assumed that mean time refers to this value of longitude. (Carlheim-Gyllensköld had

reason to believe that the longitude was in fact 10° 43' 47'' which differs by 8' 35'' and would not be consistent with GE.) GE gives the longitude of Oslo as 10° 44' 20''. Therefore, even if we consider there to have been an uncertainty in the longitude of, say, 0.1° that corresponds to an uncertainty in tidal phase lag of only about 0.1 and 0.2 deg for the diurnal and semidiurnal tides. All of these are within the phase lag measurement uncertainty for such a short record.

Our findings for harmonic constants are shown in Table 3 and the residuals of the tidal analysis are presented in Fig. 5. The latter are unremarkable except for a negative spike on 17 June which has been checked to be correctly transcribed and included in the data file. Our constants are again very similar to those of CG05.

These CG05 constants can also be found listed in Harris (1911) on page 42, station 25, but again with some misunderstanding

as to the timezone of the phase lags, hence the mistaken 'evidently erroneous' comment. Ones computed by the C&GS are shown on page 42, station 18 with local phase lags. It seems that these C&GS values were adopted by Kjær and Fjeldstad (1934), after an adjustment for the amplitude of S2 as explained on their page 20, and included in their Table 1. This set of values shown by Kjær and Fjeldstad (1934) is exactly the same as shown in the present-day ATT tables, so, although the precise history is a little obscure, it seems that they must have been adopted as the official constants for Port Virgo by the

Norwegian Hydrographic Service and passed on to the ATT. The CG05 values differ then from the resulting ATT ones only in the phase lag of S2 (leaving aside the very small O1 constituent), which seems to be an error in ATT S2 phase lag stemming back to Harris (1911).

Fjeldstad was an eminent Norwegian oceanographer and mathematician (Wikipedia, 2025f). Remarks by him on Arctic

tides and reflections on Harris's work can be found in an unpublished encyclopaedia available online from Dartmouth College (Fjeldstad, 2025).

Table 3: Amplitudes (cm) and Greenwich phase lags (deg) resulting from tidal analysis of the hourly heights at Port Virgo, Île des Danois in 1897, as reported in CG05, as given in the ATT, and as computed in the present analysis. The corresponding parameters from the FES2022b model are also shown.

| Constituent | CG05 | ATT station 895 (Danskegat) (*) | Present Analysis | FES2022b |
|---|---|---|---|---|
| M2 Amplitude | 41.4 | 41 | 41.2 | 42.33 |
| Phase lag | 15.0 | 16 | 15.8 | 20.20 |
| S2 Amplitude | 14.3 | 14 | 15.8 | 14.19 |
| Phase lag | 68.1 | 48 | 67.0 | 67.28 |
| K1 Amplitude | 2.3 | 3 | 2.8 | 5.65 |
| Phase lag | 216.4 | 204 | 207.7 | 260.06 |
| O1 Amplitude | 0.9 | 1 | 0.9 | 1.82 |
| Phase lag | 337.9 | 1 | 348.1 | 56.10 |

* The ATT lists phase lags in time zone -0100 which have been converted here to GMT.

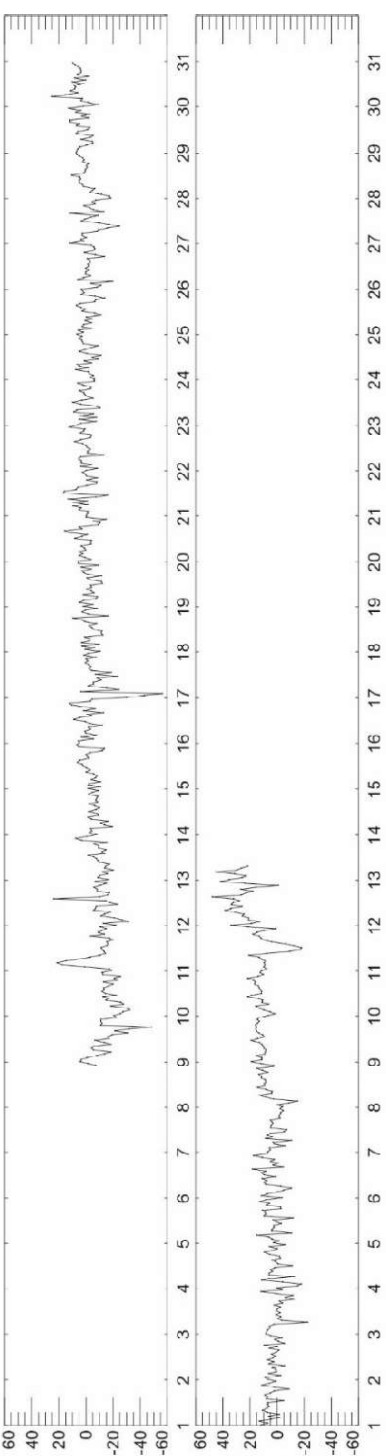

**Figure 5: Residuals of the tidal analysis for the data from Port Virgo in 1897. Units are cm.**

## 2.3 Data from Mossel Bay in 1872-73

Almost three decades before the Sorgfjord measurements discussed in Section 2.1 were made, two short sections of tide gauge record were obtained in 1872-73 from the Polhem base in Mossel Bay, only about 17 km to the west (Fig. 1b). The base was named after the Swedish scientist Christopher Polhem. Wijkander (1889) discusses these data which were acquired during the Swedish polar expedition to Svalbard led by of Adolf Erik Nordenskiöld. August Wijkander joined Nordenskiöld's expedition in 1872 but it clearly took many years for the data to be analysed.

The first short section was acquired by the crew of the brig Gladan and spanned every half hour from midnight (0 hours) on 20 October to 1630 hours on 26 November 1872 with only one interruption of 4 hours. A hole in the ice about 0.2 m square was made and through it passed a pole graduated in Swedish feet and decimal inches. (A Swedish foot comprised 10 decimal inches, the latter being 2.96 cm. It was, therefore, slightly smaller than the British foot of 30.48 cm.) One end of the pole was fixed in the sea bed and the other in the open air about 4-5 feet above the ice. A wooden float was able to rise and fall along the pole without difficulty. Measurements were made every half-hour at the 'changing of the guard' of the sailors and so, in practice, were a minute after the reported times in the Appendix of Wijkander (1889). Wijkander says he made careful adjustment for this minute in his analysis (see below).

In spring 1873, the personnel of the Polhem base made measurements similar to the Gladan ones from 0 hours on 18 February to 2100 hours on 24 April. The Gladan measurements had been made in about 5 m of water, while these 'Polhem' ones were made in shallower water approximately 2 m deep. Hourly measurements were made as exactly as possible on the hour.

Lastly, measurements of high and low tides were made from 16 May to 30 June 1873, there being not enough people available on the base to make hourly or similar measurements as before. They were made for every 5 mins for 40 mins before and after each turning point. These data were not analysed by Wijkander and we have not used them here.

The two sections of half-hourly and hourly data in 1872 and 1873 respectively were analysed by Wijkander, and have been reanalysed by us. Wijkander made a point of mentioning his adjustment for the 1-minute measurement lag mentioned above. In our opinion, this will have been irrelevant. However, much more importantly, he did not confirm in what timezone the measurements were made. They are unlikely to have been made using GMT, for example, as Greenwich was adopted as the prime meridian only in 1884. His results are listed in Harris (1911) on page 42, station 14.

For a comparison to Wijkander's values, we have assumed that the measurements were made using local mean time at the longitude of Mossel Bay and, therefore, that his reported phase lags are local ones. With this assumption, Table 4 shows that our values for M2 and S2 agree well with Wijkander, but not the phase lags of the diurnals. There was subsequently

320 found to have been a 180° analysis error in the phase lags of Wijkander's diurnal constituents (see Darwin (1889, page 560), footnote in Harris (1911, page 42, station 14), and a comment in Romagna-Manoia, 1929) which explains the differences here.

The harmonic constants for this Mossel Bay record are included in the ATT as station 892 with the name Mosselbukta. We

325 understand that they were contributed to the ATT by Russia. However, they are virtually the same as the amplitudes and phase lags listed in Fjeldstad (1936, 1939) and Hornbæk (1954), once the phase lags are adjusted to Greenwich ones from those in time zone -0100, in the case of the ATT, or from local (kappa) values, for the Fjeldstad and Hornbæk publications. Fjeldstad (1936) says that he obtained his values from Harris (1911) which are as reported by Wijkander (1889) as mentioned above. However, the problem is that, while all four Fjeldstad amplitudes are the same as the Harris ones, there

330 is a difference of about 13° in the reported phase lags of M2 and O1 (once the latter is adjusted for the 180° error for the diurnals in Wijkander's analysis mentioned in Harris (1911) and after they have all been converted to Greenwich phase lags), while those for S2 and K1 are the same. The reason for this puzzle seems to be lost in history. On the basis of our own findings (Table 4), we believe that the original Wijkander/Harris values are the more correct (once the 180° correction is made).


The residuals of our tidal analyses are shown in Supplementary Figs. 1(a,b); they contain little significant non-tidal variability.


Table 4: Amplitudes (cm) and phase lags (deg) from analysis of the Gladan and Polhem sets of data at Mossel Bay in 1872-73 by Wijkander, then his average of the two, followed by our findings for the two sets of data, and their average. We believe the Wijkander and our analysis phase lags are measured using local mean time. Therefore, we show in columns 5-7 two phase lags: one using local mean time and one with a correction to Greenwich Mean Time. The corresponding parameters from the FES2022b model (amplitude and Greenwich phase lag) are also shown.


| Constituent | Wijkander Gladan data | Wijkander Polhem data | Average Wijkander | Gladan data (this analysis) | Polhem data (this analysis) | Average (this analysis) | FES2022b |
|---|---|---|---|---|---|---|---|
| M2 Amplitude | 35 | 35 | 35 | 34.5 | 35.1 | 34.8 | 34.85 |
| Phase lag | 73 | 76 | 74 | 70.9/39.8 | 75.5/44.4 | 73.2/42.1 | 44.46 |
| S2 Amplitude | 13 | 13 | 13 | 12.8 | 13.2 | 13.0 | 11.8 |
| Phase lag | 116 | 126 | 121 | 117.8/85.7 | 124.3/92.2 | 121.1/89.0 | 92.77 |
| K1 Amplitude | 8 | 6 | 7 | 7.9 | 6.7 | 7.3 | 7.46 |
| Phase lag | 64 | 65 | 65 | 261.4/245.3 | 248.1/231.9 | 254.8/238.6 | 241.27 |
| O1 Amplitude | 3 | 2 | 3 | 3.3 | 2.2 | 2.8 | 2.16 |
| Phase lag | 243 | 235 | 239 | 65.8/50.8 | 57.3/42.4 | 61.6/46.6 | 67.77 |

**3 Comparisons to FES2022b**

FES2022b is one of several state-of-the-art models of the global ocean tide that have benefited from the availability of over three decades of near-global precise satellite altimeter information and from advances in hydrodynamic modelling (Stammer et al., 2014 ; Lyard et al., 2021; AVISO, 2025). We have used FES2022b in this paper as a source of comparison tidal constants in Tables 2-4 and as an insight into the progression of the tide around Svalbard.

Figure 6(a,b) shows the amplitude and Greenwich phase lag for M2 around Svalbard in the FES2022b model, while Supplementary Figs. 2-4(a,b) show the corresponding values for S2, K1 and O1 respectively.

Figure 6 demonstrates how M2 amplitude is lower at Mossel Bay and Sorgfjord than at Port Virgo, reducing as one travels east along the north coast of Spitsbergen, while M2 phase lag increases. A clockwise rotation of the M2 tidal wave around the archipelago is evident. The smaller S2 constituent shows similar patterns of amplitude and phase lag (Supplementary
Fig. 2). Meanwhile, the K1 diurnal tide demonstrates a more uniform distribution of amplitude and phase lag around the islands, although an amphidromic feature can be seen at the southern tip of Spitsbergen (Supplementary Fig. 3). The centimetric O1 component also exhibits comparatively uniform behaviour through the region (Supplementary Fig. 4).

Table 2 shows excellent agreement between tide gauge and model amplitudes and phase lags for all four constituents at Sorgfjord. In addition, Table 4 shows similarly good agreement for all four amplitudes and for the phase lags of M2, S2 and K1 at Mossel Bay, the latter assuming that the correction was necessary from local mean time to GMT. As anticipated from Fig. 6, M2 amplitude at Sorgfjord is slightly lower than at Mossel Bay and phase lag is about 20 degrees larger. The largest disagreement between tide gauges at the two stations and the model for any of the four constituents is a difference
of about 20 degrees in O1 phase lag at Mossel Bay. However, such a small disagreement might be expected for a centimetric constituent.

At Port Virgo (Table 3), the larger semidiurnal tides can be seen to be represented well by the model. However, there are differences in both amplitudes and phase lags for the smaller diurnal components.

Overall, one sees there is satisfactory agreement between tide gauge and model values, especially for the semidiurnal tides. In addition, this has been shown to be a good demonstration of the value of models in resolving misunderstandings in historical tide gauge data, such as in the timezone of the Mossel Bay data.

In retrospect, one can see that CG05 made an error in being reassured by the apparent similarity in M2 amplitude and (to a lesser extent) phase lag reported by Wijkander for Mossel Bay, as a confirmation of his own findings at Sorgfjord. It seems that Carlheim-Gyllensköld failed to realise that the phase lags reported by Wijkander used local mean time, while his own were expressed using GMT. This error was largely compensated for in his calculation by the ~20 deg difference in real phase lag between the two locations (Fig. 6b), which of course he would not have known about. Similarly, he would

not have been aware of the eastward reduction in M2 amplitude along the north coast of Svalbard which we now know from models.

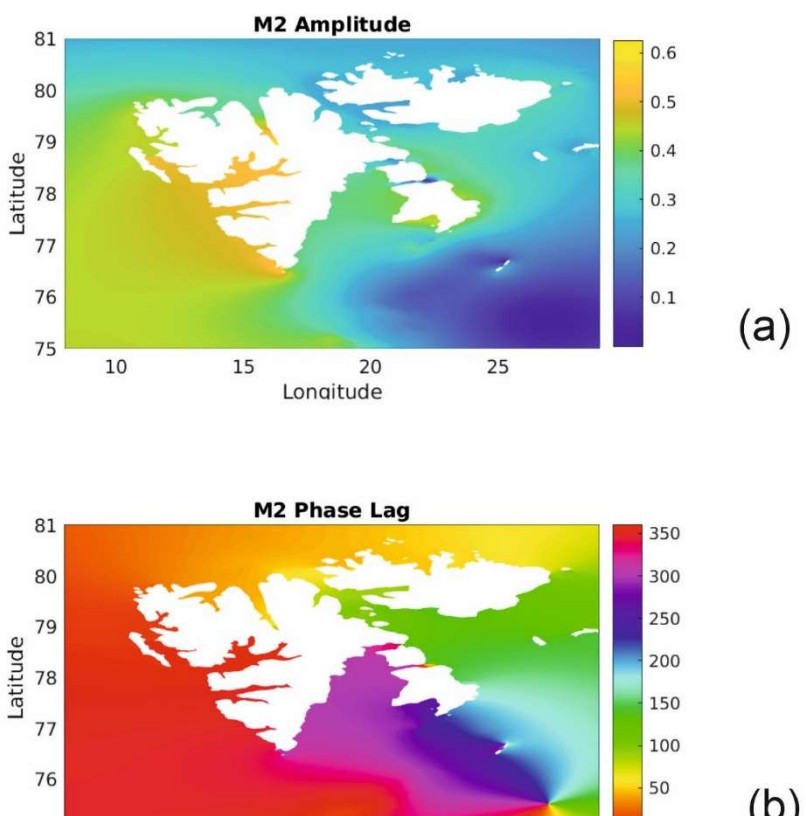

**Figure 6: (a) Amplitude (m) and (b) Greenwich phase lag (deg) for M2 around Svalbard from the FES2022b model.**

## 4 Conclusions

This report has discussed three historical tide gauge records from the high latitude of Svalbard. The first record was obtained in 1900 by measuring variations in the height of the floating ice near to the Swedish winter base at Sorgfjord. Tide gauges of various kinds on floating ice were used before that of Jäderin (e.g. Ross, 1854) and were used afterwards, for example during the Australasian Antarctic Expedition in 1911-14 (e.g. Doodson, 1939) and the Oxford University Arctic Expedition to Brandy Bay in north-east Svalbard in 1936 (Fjeldstad, 1939). A particularly impressive time series of hourly tidal heights

was obtained close to the Bering Strait during the famous Vega Expedition to the Russian Arctic coast led by Nordenskiöld. This was obtained using a cable passing through a hole in the pack ice and spanned 7 December 1878 to 7 June 1879. The data were analysed by Scherneck and Ekman (1999).

However, Jäderin's is the only example known to us of this particular type of tide gauge. It does, therefore, deserve to be

better known in the history of tide gauge recording. At first sight, it is surprising that such a crude design worked at all, but in fact it did work well for over 3 months as evidenced by the quality of the tidal analysis of its data.

We understand from the Norwegian Hydrographic Service that there have been two more recent tide gauge recordings in Sorgfjord: for one week in 1938 and for three weeks in 2014. However, both of these records are much shorter than the

1900 data set, and in our opinion a more meaningful comparison to the present-day tide can be made using a state-of-the-art numerical global tidal model. Table 2 shows that the harmonic constants obtained from the 1900 record agree well with those from FES2022b, suggesting that there have been no large changes in the tide in that part of Svalbard in the last century.

The second record was obtained from Port Virgo, Île des Danois (Danskegat) in 1897 by conventional reading of a tide board, and so has less technical novelty. Nevertheless, the sparse coverage of tide gauge recording in the Arctic, even nowadays, means that it has some scientific value. Again, more recent measurements have been made at this location: for almost two weeks in both 2008 and 2009. However, for comparison purposes, we have once again made use of the FES2022b model which suggests no major changes, in the semidiurnals at least, over the last 100 years.


Finally, we have analysed two short sections of data from Mossel Bay obtained earlier in 1872-73. The location of these measurements was near to Sorgfjord, and so it is unsurprising that the tidal constants obtained are roughly similar (allowing for assumptions on the timing of the Wijkander data, see above). Once again, they are in decent agreement with FES2022b.

We know that other short tide gauge records were acquired from Svalbard during the late 19[th] century. For example, Ekholm (1891) mentions measurements for about a week in 1873 during another polar Swedish expedition, to Cap Thordsen on the

west coast of Spitsbergen. Recording was expanded considerably from the mid-20[th] century: Hornbæk (1954) lists harmonic constants from 15 locations in Svalbard. He also provides constants from two locations in Jan Mayen and three in East Greenland. Kjær and Fjeldstad (1934) may also be referred to, listing constants from measurements at Bear Island and East Greenland in the early 1930s.

One particular measurement campaign to refer to was that of the Italian Hydrographic Institute which established a tide gauge station in 1928 at the 'London' base in Kings Bay (Kongsfjorden) about 4.5 km north of Ny-Ålesund on the island of Blomstrandhalvøya (Romagna-Manoia, 1929). This work was carried out as part of navy support to Umberto Nobile's airship flights to the North Pole (Wikipedia, 2025g). Tabulations of hourly values of sea level from 6 June to 3 August are given by Tenani (1939) while similar sets of harmonic constants from that record are listed by Romagna-Manoia (1929), Tenani (1939) and Hornbæk (1954). The values given by Romagna-Manoia (1929) appear identical to those quoted for Kongsfjorden in the ATT Tables (station 898); this would be another example of century-old information being included in modern data banks. As expected, the constants are also virtually the same as those for Ny-Ålesund in the ATT (station 898a). However, an important point to make concerning this particular example is that several tide gauge benchmarks are said to have been established on the island and one on the south beach of Ny-Ålesund. Therefore, if the marks survive and if a modern set of measurements could be made at the same location, then long-term changes in sea level in the area could be investigated as well as changes in the tides.

A concluding general point to make about most of the polar tidal measurements we have investigated is that they appear to have been made by dedicated, and often highly qualified, participants who took the trouble to obtain good data. We therefore believe that the 'data archaeology' of historical information such as described in this report can potentially benefit present-day research.

**Author Contributions**

The existence of the tide gauge tabulations in CG05 was originally discovered by TA. They and the Mossel Bay tabulations were converted to computer files by PLW who undertook the tidal analyses. Both authors contributed to the construction of the manuscript.

**Data and Code Availability**

All data discussed in this article are included provided in the Supplementary Material. A version of the TASK tidal analysis code may be obtained from the Innovations Group of the National Oceanography Centre.

**Competing Interests**

The authors declare that they have no competing interests.

**Acknowledgements**

Chris Jones of the UK Hydrographic Office and Greg Dusek and Todd Ehret from NOAA are thanked for researching the origin of the harmonic constants in the ATT. Oda Roaldsdotter Ravndal and Hilde Sande Borck of the Norwegian Mapping Authority provided information on tide gauge recording and tidal modelling around Svalbard. Thomas Hammarklint of the Swedish Maritime Administration provided background to the Petrelius tide gauge. The Wijkander article was provided by Maria Asp of the Center for History of Science, The Royal Swedish Academy of Sciences, while Fjeldstad (1936) was provided by Marianne Bjordal Strømsholm of the University of Bergen Library. We are grateful to Loren Carrère from Collecte Localisation Satellites (CLS) and Florent Lyard of the Laboratoire d'Etudes en Géophysique et Océanographie Spatiales (LEGOS) for the FES2022b model values. We thank both reviewers for their valuable comments.

**Supplementary Material**

There are two files in the supplementary material containing the hourly values for the stations discussed in CG05: 'data_baie_de_treurenberg' and 'data_portvirgo' respectively. There are also two files containing half-hourly and hourly data from Mossel Bay in 1872-73: 'data_wijkander1' and 'data_wijkander2'. Each data file contains 20 lines of header information which describes their format. There is also a file 'smfigs.pdf' which shows the tidal residuals for the two Mossel Bay records and distributions of the S2, K1 and O1 constituents around Svalbard from FES2022b.

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
