# Peer review of "Three Historic Tide Gauge Records from Svalbard"

_History of Geo- and Space Sciences, 2025_

## Author Response (AR1)

We already provided the editor and reviewers with our final responses on 1 December. Shown below again.

1 December 2025

Dear Editor,

Many thanks for the responses of the reviewers and for your comments. We intend to include most of them in a new version of the paper in addition to some modifications of our own. We will also attend to a request of the editorial office for additions to the captions of Figures 2 and 3 and for a 'to be submitted to' reference to be replaced.

Our responses to the comments of the reviewers are as follows in capitals:

Reviewer 1:

Paragraph 145: Just a comment regarding harmonic constants from NOAA sources; these sometimes refer the phase lags as 'kappa', i.e. the local equilibrium tide meridian rather than 'g', the Greenwich meridian. This could be a source of difference in any datasets sourced from NOAA or the US Coast & Geodetic Survey.

YES, THAT IS RIGHT, BUT WE BELIEVE WE UNDERSTAND THINGS NOW. THE HARRIS PAPER CONTAINS AMPLITUDES AND KAPPA PHASE LAGS FOR THE TREURENBERG/SORGFJORD AND PORT VIRGO/DANSKEGAT STATIONS. THE AMPLITUDES ARE EXACTLY THE SAME AS IN THE ATT, BUT THE KAPPA PHASE LAGS ARE SLIGHTLY DIFFERENT TO THE ATT ONES SAID TO BE IN -0100. BUT, IF YOU CONVERT THEM BOTH TO GREEENWICH ONES, YOU GET THE SAME THING (WITHIN THE PRECISION OF THE ARITHMETIC OF COURSE.)

WE DID NOT IN FACT GET ANYTHING FROM NOAA SOURCES, ONLY FROM THE ATT WHICH WE UNDERSTOOD TO HAVE BEEN CONTRIBUTED TO THE ATT BY NOAA. IT SEEMS TO US THEREFORE THAT NOAA MUST HAVE KNOWN WHAT THEY WERE DOING AND CONVERTED THE KAPPAS TO -0100 VALUES BEFORE PROVIDING THEM.

Tables 1, 2 and 3; just a suggestion; give all phase lags as three values of degrees, i.e. 64.6° show as 064.6° etc. Maybe also add in column 1, alongside each constituent name, the titles of 'amplitude' and 'phase' to make it clearer which are the values in each row ....

WE DON'T SEE THE POINT OF HAVING PHASE LAGS WITH A LEADING ZERO. BUT WE AGREE WITH THE SUGGESTION OF ADDING AMPLITUDE AND PHASE LAG IN THE FIRST COLUMN OF THE TABLES.

Paragraph 180; might a schematic diagram of the second-stilling well type of device be helpful to visualise the set-up?

WE DO NOT HAVE DRAWINGS OF THE ACTUAL STILLING-WELL FLOAT GAUGES USED IN SORGFJORD, BUT WE HAVE PUT IN THE TEXT A MENTION THAT PUGH AND WOODWORTH (2014) CONTAINS A DRAWING OF A TYPICAL FLOAT GAUGE.

Paragraph 205: Suggest the sentence "The second record discussed in CG05 is a short one spanning 2300 hours on 8 June to 800 hours on 18 July 1897....." be re-written as follows:

""The second record discussed in CG05 is of shorter duration, commencing at 2300 hours on 8 June to 0800 hours on 18 July 1897....."

DONE

[and suggest any times are written in 24 hour notation throughout the paper, i.e. as in the '0800' shown above].

THERE ARE NO OTHER EXAMPLES LIKE THIS TO BE CHANGED. WE HAVE LEFT MIDNIGHT AS 0 HOURS RATHER THAN 0000 HOURS.

Also it could be helpful to include the time interval of these readings at this stage of the text (i.e. hourly values) - this information is provided but not until paragraph 220.

REWORDED AS SUGGESTED

Paragraphs 235/240; Regarding the coordinate datum assumptions - given the original gauge was 1800s, this was of course long before GNSS and therefore WGS84, whereas the coordinates are related to Google Earth positions. It's worth noting as an additional uncertainty, even though the impact would be minimal.

THE UNCERTAINTIES IN LONGITUDE, AND SO THE CONVERSION OF PHASE PAGS TO GREENWICH ONES, ARE COVERED BY THE 0.1 DEG UNCERTAINTY MENTIONED IN THE TEXT.

Paragraph 255; just a question; is the ATT S2 phase lag considered incorrect for ATT port number 0895 Danskegat? (i.e. 078° in timezone -0100, converted to 048° in GMT). If so, should it be amended to 068.1° GMT (i.e. 098.1° in timezone -0100)?

THAT'S RIGHT. PORT VIRGO/DANSKEGAT IS STATION 18 IN THE HARRIS PAGE 42 TABLE WITH AN AMPLITUDE OF 0.26 FEET (8 CM) AND A PHASE LAG (KAPPA) OF 70 DEG. FOR THE REASONS WE EXPLAINED TO DO WITH KJAER AND FJELDSTAD THE AMPLITUDE WAS JACKED UP TO 14.3 CM BUT THE PHASE WAS UNCHANGED. THAT KAPPA OF 70 CORRESPONDS TO THE 78 DEG FOR g IN -0100 IN THE ATT AND TO 48 DEG GREENWICH. SO, AS WE BELIEVE THAT THE 48 SHOULD BE MORE LIKE 67.0 (OR 68.1 IF YOU WANT TO USE THE CG05 NUMBER), WE BELIEVE THE ORIGINAL HARRIS VALUE WAS TOO LOW BY ABOUT 19 OR 20 DEG, AND ALSO THEREFORE THE ATT VALUE. THIS ERROR THEREFORE GOES BACK TO HARRIS A CENTURY AGO.

Paragraph 285: it might be worth mentioning the difference between a Swedish foot and a 'normal' foot.

EXTRA WORDING ADDED.

Paragraph 295: the phrases 'flux' and reflux'; presume these align to High and Low Water respectively? And if so, should this be pointed out?

PLEASE SEE THE REPLY TO THE SAME COMMENT BY REVIEWER 2.

Paragraph 365 Figures (a) and (b); would it be helpful to show the positions of the study areas on these maps of the M2 phase lag and amplitude?

WE HAVEN'T DONE THIS. IT WOULD MAKE FOR CLUTTER IN WHAT WE ARE TRYING TO SHOW ABOUT THE MODEL IN THE SMALL FIGURES. IT IS EASY FOR ANYONE WHO WISHES TO TO COMPARE THESE FIGURES TO FIGURE 1.

Reviewer 2:

The article might benefit from a summarizing table of the three tide gauge locations, to give the reader a quick overview of the different campaigns. This table could include the location of the measurement, the date and the length of the data series, as well as the tide gauge type if relevant. The table could for instance be placed in relation to Figure 1, or in another location the authors find appropriate.

AN EXTRA TABLE HAS BEEN ADDED AS SUGGESTED.

In Figure 1, a suggestion would be to write out the names of the locations instead of only the first letter. This would improve the readability of the map. In addition, I would suggest including a map showing the location of Svalbard in the Arctic - similar to the map shown by the Norwegian Polar institute in their overview on https://toposvalbard.npolar.no for the reader to more easily situate the archipelago and understand the specific challenges the different expeditions mentioned faced.

WE HAVE NOT PUT THE NAMES IN FULL AS WE THOUGHT IT WOULD MAKE THE MAP TOO CROWDED ESPECIALLY AS IT IS LIKELY TO BE REDUCED IN SIZE WHEN PRINTED. ALSO WE HAVE ADDED A COUPLE OF OTHER LOCATIONS TO IT. WE HAVE ADDED A LARGER SCALE MAP AS SUGGESTED AS FIGURE 1(A).

Comment related to Tables 1-3. For readers not familiar with tidal constituents, it can be difficult to understand the differences between amplitudes and phases as they are recorded in the tables. I would suggest to separate the amplitude and phases in two columns, or in another way making the difference clearer to the reader.

WE HAVE ADOPTED THE SUGGESTION OF REVIEWER 1 AND HAVE ADDED 'AMPLITUDE' AND 'PHASE LAG' ON EACH LINE. IT SHOULD BE CLEAR NOW.

Line 262: There is a misspelling of the name Harris.

FIXED

Line 282: I suggest that for readability, the sentence "The base was named after..." is moved up and is placed after the first sentence in the paragraph, after "from the Polhem base in Mossel Bay, only about 17km to the west (Fig. 1)."

DONE

Line 297: The words "flux" and "reflux" I assume, based on the context, are referring to high and low tide. As these are not commonly used words in this context, it would be worth explaining them to the reader.

THANKS FOR MENTIONING THIS. THESE TWO WORDS WERE USED BY WIJKANDER AND WE COPIED THEM (INCORRECTLY IN RETROSPECT) FROM HIS PAPER. 'FLUX' AND 'REFLUX' NORMALLY REFER TO THE TIMES OF RISING AND FALLING TIDE AND NOT TO THE ACTUAL HIGH AND LOW WATERS THEMSELVES. HOWEVER WIJKANDER CLEARLY MEANT THE LATTER, AS ONE CAN UNDERSTAND FROM HIS TEXT A FEW LINES LATER. WE HAVE CHANGED OUR OWN TEXT TO REMOVE THESE TWO WORDS.

Paragraph 382-387: Just a note, the Norwegian Hydrographic Service has also measured the water level in the Mossel Bay. Both the series from Mossel Bay and Sorgfjorden have been analysed, and although the series are short, they show similar constituents as the ones described here.

OK THANKS